# Differential Effects of Task-Irrelevant Monaural and Binaural Classroom Scenarios on Children’s and Adults’ Speech Perception, Listening Comprehension, and Visual–Verbal Short-Term Memory

**DOI:** 10.3390/ijerph192315998

**Published:** 2022-11-30

**Authors:** Larissa Leist, Carolin Breuer, Manuj Yadav, Stephan Fremerey, Janina Fels, Alexander Raake, Thomas Lachmann, Sabine J. Schlittmeier, Maria Klatte

**Affiliations:** 1Cognitive and Developmental Psychology Unit, Center for Cognitive Science, Department of Cognitive Psychology, University of Kaiserslautern-Landau, 67663 Kaiserslautern, Germany; 2Institute for Hearing Technology and Acoustics, RWTH Aachen University, 52074 Aachen, Germany; 3Audiovisual Technology Group, Technische Universität Ilmenau, 98693 Ilmenau, Germany; 4Centro de Investigación Nebrija en Cognición, Facultad de Lenguas y Educacion, Universidad Nebrija, 28015 Madrid, Spain; 5Teaching and Research Area of Work and Engineering Psychology, RWTH Aachen University, 52066 Aachen, Germany

**Keywords:** auditory distraction, children, speech perception, listening comprehension, verbal short-term memory, irrelevant sound effect, binaural, monaural, classroom, learning

## Abstract

Most studies investigating the effects of environmental noise on children’s cognitive performance examine the impact of monaural noise (i.e., same signal to both ears), oversimplifying multiple aspects of binaural hearing (i.e., adequately reproducing interaural differences and spatial information). In the current study, the effects of a realistic classroom-noise scenario presented either monaurally or binaurally on tasks requiring processing of auditory and visually presented information were analyzed in children and adults. In Experiment 1, across age groups, word identification was more impaired by monaural than by binaural classroom noise, whereas listening comprehension (acting out oral instructions) was equally impaired in both noise conditions. In both tasks, children were more affected than adults. Disturbance ratings were unrelated to the actual performance decrements. Experiment 2 revealed detrimental effects of classroom noise on short-term memory (serial recall of words presented pictorially), which did not differ with age or presentation mode (monaural vs. binaural). The present results add to the evidence for detrimental effects of noise on speech perception and cognitive performance, and their interactions with age, using a realistic classroom-noise scenario. Binaural simulations of real-world auditory environments can improve the external validity of studies on the impact of noise on children’s and adults’ learning.

## 1. Introduction

Learning in classrooms is often impeded by unfavorable acoustic conditions, such as noise and reverberation [1]. A recent study in German preschool and school classrooms reported an average sound pressure level (SPL) of 66 dB L_A,eq_ (A-weighted equivalent continuous SPL), and a range of 62–69 dB L_A,eq_ during typical activities [2]. Other studies conducted across Europe and the US reported values ranging from 42 to 100 dB L_A,eq_ [3,4] (see also Table 1 in [2]). Field studies revealed that children instructed in classrooms with high levels of indoor or external (aircraft) noise score lower in achievement tests and in ratings of well-being at school, and exhibit higher levels of annoyance due to noise [5,6,7,8,9]. Numerous experimental studies have analyzed effects of acute noise on children’s performance in a range of auditory and non-auditory tasks. Concerning the former, it has consistently been shown that children’s language comprehension is more impaired than adults’ by noise and reverberation [10,11,12,13]. Concerning non-auditory tasks, findings are less consistent for complex academic tasks such as reading and numeracy [14,15,16] (for review, see [17], but reliable noise-induced performance decrements have been reported for children’s visual–verbal short-term memory [18,19,20,21].

However, most of these studies do not represent real-world auditory environments with respect to the type of noise and its perception by the person affected. For example, the noise maskers used in many psychoacoustic studies on speech-in-noise perception have nothing in common with a noisy classroom. In addition, in the vast majority of studies on noise effects on cognitive performance, the noise is presented monaurally (the same signal presented to both ears) over headphones, and in cases of loudspeaker presentation the same signal is sent to each speaker. These presentation formats oversimplify the multiple features of binaural hearing in complex acoustic environments, where sounds are spatially spread across the room, and sound sources change often and unpredictably. In order to represent such complex scenes in laboratory settings, one approach is binaural reproduction, wherein the interaural differences in sound reaching the ears are authentically represented, including spatial cues [22,23]. In the current study, we analyzed the effects of a realistic classroom-noise scenario on tasks requiring processing of auditory (Exp. 1) and visual information (Exp. 2) in children and adults. We aimed to assess the detrimental noise effects across age groups, and to explore whether and to what extent these effects, and the developmental change associated with them, are moderated when a realistic, binaural presentation mode is used instead of the usual, monaural presentation.

## 2. Experiment 1: Effects of Classroom Noise on Speech Perception and Listening Comprehension

Learning in classrooms relies heavily on oral instruction and listening in the presence of irrelevant sounds. Thus, school children are regularly faced with the requirement of focusing on a specific sound source while ignoring others [24]. Experimental studies on the effects of environmental noise on children’s ability to understand speech have focused on simple speech perception tasks requiring identification of isolated speech targets in noise and/or reverberation. However, listening requirements faced by children in classrooms go far beyond pure identification. Effective listening in these situations requires storage and processing of complex oral information in working memory, while constructing a coherent mental model of the information presented [25]. There is evidence that noise may affect the storage and processing of spoken items even when the signal-to-noise ratio (SNR; signal SPL minus noise SPL) is high enough to allow perfect or near-perfect identification [26,27,28]. Thus, effects of noise on word identification tasks do not allow predictions of decrements in complex listening tasks.

Studying the impacts of noise and reverberation on children’s speech perception in a classroom-like setting, Klatte and colleagues [29] found differential effects of foreign, single-talker speech and classroom noise without speech on word identification (word-to-picture matching) and listening comprehension (acting-out complex oral instructions). In both tasks, children were affected more than adults. In the comprehension task, both speech and classroom noise significantly reduced children’s performance, with 6- to 7-year-old first-graders suffering the most, whereas adults were unaffected. Speech was more disruptive than classroom noise. In contrast, word identification was much more impaired by classroom noise when compared to speech. The authors proposed that, with the SNRs of −3 dB to 3 dB used in their study, the effects of background speech and classroom noise resulted from different mechanisms. Classroom noise masked the speech targets. Background speech was a less potential masker, but interfered with short-term memory processes that children (but not adults) rely on when listening to complex sentences. The study further revealed that the children’s ratings of the sound-induced disruption were unrelated to their objective performance decline. This finding underlines that, while it is undeniable that noise has a negative impact on performance, this does not mean that a person affected is subjectively aware of these effects or feels annoyed; cf. [30].

In Klatte et al. [29], the background sounds were presented via loudspeakers located at the sides of the laboratory room. The same recording was sent to each of the eight loudspeakers, and the target signals were presented through a separate speaker located in front of the room. In the current study, we further increased the realism of the design by Klatte and colleagues [29], by including a classroom-noise scenario that is reproduced binaurally (i.e., authentically representing interaural and spatial cues). Prior studies confirmed that participants’ performance in listening tasks is affected when the realism of the auditory scene is increased [31,32,33,34].

### 2.1. Materials and Methods

Participants: The sample consisted of 36 student volunteers (19 female) from the University of Kaiserslautern, aged between 19 and 31 years (*M* = 24.9, *SD* = 3.9 years), and 56 children recruited from a primary school in Kaiserslautern. The child sample comprised 37 second-graders (9 female), aged between 6 years, 3 months and 8 years, 2 months (*M* = 7 years, 5 months, *SD* = 3 months); and 19 third-graders (12 female) aged between 8 years, 3 months and 9 years, 7 months (*M* = 8 years, 9 months, *SD* = 4 months).

All participants were native German speakers and had normal or corrected-to normal vision and normal hearing according to either self-reports (adults) or parental reports (children). The study was approved by the Rhineland-Palatinate school authority and by the Ethics Committee of the University of Kaiserslautern. Informed written consent was provided by the adult participants and by the children’s parents. Adults received either course credit or payment for participation (10 €).

Apparatus: The tasks were created in Python 3.7/PsychoPy 3.1.5 [35] and executed using a 15.6-inch laptop (HP ProBook 450 G6). The screen’s resolution was 1920 × 1080 pixels, and its refresh rate was 60 Hz. The sounds were presented via Sennheiser HD650 headphones and a Focusrite Scarlett 2i2 2nd generation audio interface.

Speech signals: The words and the instructions were read by a professional female speaker in a sound-attenuated booth and recorded with a Sennheiser MD 421-II-4 Dynamic Studio Microphone at a sampling rate of 44.1 kHz and 16-bit resolution. The recordings were loudness-normalized according to EBU R-128 [36] using Version 3.0.0 of Audacity^®^ recording and editing software [37].

Classroom-noise scenario: The background noise represented a classroom-like auditory environment with sounds from everyday classroom activities, e.g., furniture use; desk noise, including writing and use of other stationary items; footsteps; door opening and closing; use of zippers on bags; undoing a plastic wrapper; and turning the pages of a book. To prevent any learning effects, the different noises were presented at irregular intervals as in real classroom scenarios. Some noises (e.g., writing) were played more often than others (e.g., door) to mimic their typical frequency of occurrence in reality. The selection of nonspeech sounds and the frequency with which they occurred were based on listening to recordings of lessons in medium-sized classrooms. Pink noise (−5 dB/octave decay slope) presented at *L*_Aeq,1m_ of 41.5 dB provided a steady-state noise that simulated air-conditioning noise throughout the auditory scene.

As children frequently speak in class, multi-talker speech, consisting of four child voices talking in Hindi, which was foreign to all participants, was added to the scene. We recorded (32-bit, 44.1 kHz sampling rate) a child (8 years old) having a natural, unscripted conversation with an adult (the parent) on a range of topics for around two hours in a hemi-anechoic room. The talkers sat on chairs facing each other at a 2 m apart. Each talker wore a DPA 4066 omnidirectional headset microphone positioned 7 cm from the center of lips, as was done in previous research [38]. This recording was post-processed to remove all the adult speech parts, silences, and other artefacts. The remaining speech was segmented into smaller sentences, and the fundamental frequencies of 3/4 sentences were changed. Hence, speech from 4 child voices was created for use in the auditory scene. Individual components of the speech segments were only presented once. In the final auditory scene, two child voices were active at any time. Following a previous study [38], the order of active talkers changed randomly.

To auralize the binaural background sounds, a classroom model was generated in SketchUp [39]. The classroom has a rectangular floor plate and flat ceiling (9 m × 11 m × 2.7 m, W × L × H), and all the room surfaces were assigned an absorption coefficient of 1 (i.e., anechoic conditions). Within the room, 12 desks and chairs were modeled in four rows. Each chair seated a child. Typical sound absorption values were assigned to each child. The listener’s position referred to a child sitting in the middle of the last row with an ear height of 1 m. A total of 16 different sound sources were then placed in the room. Twelve sound sources for non-speech sounds were placed at varying distances around the listener; there were equal numbers of sound sources on the left and right sides. The directionality of the last four sound sources represented two child talkers at 3 m away at ± 30 degrees, and two talkers at 5 m away at ±20 degrees, with an ear height of 1 m. The classroom model was auralized using RAVEN [40]. A generic head-related transfer function (HRTF) from the FABIAN dummy head [41] with a resolution of 1° × 1° was used. Although it is well known that HRTFs differ significantly between adults and children [42], and that this difference can influence cognitive tasks [31], a generic solution of HRTFs was used in the current study, as this was the first attempt to spatially separate the sounds. Thus, in the binaural condition, the sounds were spatially spread across the room, and the order of active talkers and source locations of non-speech sounds changed randomly, as is typical in real classrooms. In the monaural condition, the sounds were presented without any spatial separation, and appeared to come from straight ahead (or inside the head) to the listener.

For both sound conditions, the auralized files were mixed down to a 2-channel audio file. The headphone (Sennheiser HD 650, Wedemark, Germany) output per channel was calibrated using Brüel and Kjær Artificial Ear Type 4153 with a Brüel and Kjær Type 4190 omnidirectional microphone capsule.

For both sound conditions, the *L*_A,eq_ of the target speech signals and the classroom noise were 60 and 63 dB, yielding a SNR of −3 dB [29].

Tasks: We used modified versions of the tasks from Klatte and colleagues [29]. As we included three sound conditions (silence, monaural noise, binaural noise), three equivalent, parallel versions of each task were constructed.

Speech Perception: A word-to-picture matching task requiring discrimination between phonologically similar words was used to measure speech perception. A total of 84 lists of four phonologically similar German nouns (for example, Kopf (head), Topf (pot), Knopf (button), and Zopf (braid)) were constructed. Each word was represented by a simple and easy-to-name colored drawing. Each trial began with a 1.5 s visual cue, followed by a spoken word. Then, the screen displayed four images in a fixed array, one representing the target word and three representing similar-sounding distractor words (see Figure 1). The position of the picture representing the target words was counterbalanced. The participant’s task was to mouse-click on the picture that corresponded to the target word. There were 28 trials in each sound condition. The task was the same for both adults and children.

Listening comprehension: A paper and pencil test requiring the execution of complex oral instructions was used to assess listening comprehension. In each of the sound conditions, participants heard 8 oral instructions, such as “Male ein Kreuz unter das Buch, das neben einem Stuhl liegt” (“Draw a cross under the book that is next to the chair”). The task was to carry out the instructions on pre-prepared response sheets. On the response sheets, each instruction was represented by a row of small black-and-white drawings of the target objects (e.g., a book next to a chair) and distractor stimuli (e.g., a book next to a ball) (see Figure 2). The response sheet was also visible on the computer screen in front of the participant. A red arrow indicated the row reflecting the current instruction on the response sheet. Each instruction began with an auditory cue (bell ringing). Participants had 18 s after the end of an instruction to complete the entries on the response sheet. They were instructed to begin carrying out the instructions as soon as feasible. Scoring was based on the number of elements correctly executed according to the respective instructions. In order to equalize task difficulty across age groups, the adults received longer and syntactically more complex instructions.

Subjective disturbance was assessed using a smiley scale with 4 points representing the ratings “not at all disturbed” (0), “a little disturbed” (1), “strongly disturbed” (2), or “extremely disturbed” (3).

Procedure: Both children and adults were tested in separate groups of 2 to 4 in a sound-attenuated booth at the University of Kaiserslautern-Landau. Four computer workplaces were arranged in the room, with about 4 m and partition walls between them. The walls around each workstation were equipped with posters of a primary school classroom to create a more classroom-like atmosphere. Adults received written instructions. Children were instructed orally by a researcher. Participants were informed that they should ignore the sounds and focus solely on the execution of the respective task. The experimenter stayed in the back of the laboratory room during the whole session.

Each participant performed both tasks in each of the three sound conditions (silent control, monaural, and binaural auditory classroom scene). Sound conditions were varied in blocks of trials. There were 28 and 8 trials per block in the word identification and listening comprehension tasks, respectively. The order of sound conditions and the allocation of test versions to sound conditions were counterbalanced between participants.

Each session started with a general instruction provided by the experimenter, followed by the monaural presentation of the classroom scene for 4 s to familiarize the participants with the background sound. Then, all the pictures used in the speech perception task were presented and named. Subsequently, participants performed the speech perception task. Thereafter, the listening comprehension task was instructed and performed. Both tasks started with four practice trials. In the sound conditions, the classroom-noise scenario was continuously played during the respective block of trials. Finally, the monaural and binaural auditory scenes were played for 10 s in order to complete the disturbance ratings. The session took about 40 min in total.

### 2.2. Results

Mean proportion correct scores and standard deviation as a function of task, sound condition, and age group are depicted in Table 1. Difference scores were calculated for each participant by subtracting proportion correct scores in noise from performance in silence. These scores were used as dependent variables. Figure 3 depicts the mean difference scores with respect to task, sound condition, and age group.

**Table 1 ijerph-19-15998-t001:** Mean proportion correct scores for speech perception and listening comprehension, and mean disturbance ratings for Experiment 1 as a function of sound condition and age group (standard deviation in parenthesis).

Task/Ratings	Sound Condition	Adults	2nd Graders	3rd Graders
		*M* (*SD*)	*M* (*SD*)	*M* (*SD*)
Speech Perception	Silence	0.99 (0.00)	0.97 (0.05)	0.96 (0.06)
	Monaural	0.79 (0.05)	0.50 (0.13)	0.61 (0.13)
	Binaural	0.92 (0.04)	0.74 (0.11)	0.77 (0.11)
Listening comprehension	Silence	0.85 (0.10)	0.94 (0.01)	0.94 (0.08)
	Monaural	0.79 (0.12)	0.76 (0.15)	0.85 (0.12)
	Binaural	0.84 (0.11)	0.76 (0.14)	0.87 (0.07)
Disturbance rating	Monaural	1.25 (0.55)	0.97 (0.93)	1.11 (0.66)
	Binaural	2.00 (0.72)	1.35 (0.98)	1.53 (0.84)

For speech perception, a 3 *×* 2 mixed ANOVA of the difference scores with age group (adults, second-graders, third-graders) as a between-subjects factor and sound condition (monaural, binaural) as a within-subject factor confirmed a significant main effect of the sound condition—*F*(1, 89) = 222.31, *p* < 0.001, *partial η*^2^ = 0.714—reflecting stronger impairment in the monaural when compared to the binaural condition; a significant main effect of age group—*F*(2, 89) = 56.29, *p* < 0.001, *partial η*^2^ = 0.558—reflecting stronger impairments in the children when compared to adults; and a significant interaction—*F*(2, 89) = 9.19, *p* < 0.001, *partial η*^2^ = 0.171. Post hoc *t*-tests revealed that, in each of the age groups, the ability to recognize isolated words was less impaired in the binaural condition when compared to the monaural condition (all *p’s* < 0.001). The interaction reflects a more pronounced difference between age groups in the monaural when compared to the binaural noise condition (see Figure 1). Separate analyses per sound condition revealed significant differences between all age groups for monaural noise, and significant differences between adults and both groups of children for binaural noise (all *p’s* < 0.01), whereas second- and third-graders did not differ (*p* = 0.190).

For listening comprehension, the 3 *×* 2 mixed ANOVA revealed a significant main effect of age group—*F*(2, 89) = 24.87, *p* < 0.001, *partial η*^2^ = 0.359. The effect of sound condition and the interaction were not significant (*F*(1, 89) = 2.67, *p* = 0.106, *partial η*^2^ = 0.029 and *F*(2, 89) = 1.17, *p* = 0.314, *partial η*^2^ = 0.026). Concerning the main effect of age, post hoc tests revealed that second-graders were more impaired than adults (*p* < 0.001) and third-graders (*p* < 0.001). No significant differences were found between third-graders and adults (*p* = 0.27). One-sample *t*-tests revealed that the performance decrement due to binaural noise in adults did not differ significantly from 0; *t*(35) < 1.

In a further step, we analyzed whether speech perception in noise predicts listening comprehension in noise. In view of the small sample size for the third-graders and the fact that the adults’ listening comprehension was largely unaffected by noise, the respective correlation analyses was confined to the second-graders. Correlations between proportion correct scores for speech perception and listening comprehension in noise were calculated. In the binaural condition, speech perception was significantly related to listening comprehension, *r* (35) = 0.375, *p* < 0.05, whereas in the monaural condition, speech perception and listening comprehension were unrelated (*p* = 0.19).

For the disturbance ratings, the 3 *×* 2 mixed ANOVA yielded a significant main effect of noise condition—*F*(2, 89) = 33.10, *p* < 0.001, *partial η*^2^ = 0.271—reflecting higher disturbance in the binaural when compared to the monaural condition; a significant main effect of age group—*F*(2, 89) = 4.20, *p* < 0.05, *partial η*^2^ = 0.09—but no interaction: *F*(2, 89) = 2.09, *p* = 0.13. Post hoc tests confirmed that the disturbance ratings were higher in adults when compared to second-graders (*p* < 0.05). Ratings of the third-graders did not differ from those of the second-graders and adults (*p* = 1 and *p* = 0.35, respectively). Further analyses in the second-graders and adults confirmed that, for both noise conditions, the disturbance ratings were unrelated to word identification and listening comprehension in noise (proportion correct scores) and unrelated to the actual performance decrements (difference scores) in adults (all *p’s* > 0.10) and children (all *p’s* > 0.40).

### 2.3. Discussion

Experiment 1 replicated the often-reported finding that children are less able than adults to understand speech in the presence of background noise [10,11,12,13,29]. In the monaural condition, word identification was more impaired in second-graders when compared to third-graders, and more impaired in third-graders than in adults. However, in each of the age groups, word identification performance was substantially less affected with binaural noise when compared to monaural presentation of the classroom-noise scenario. This indicates that children and adults may use the spatial cues inherent in the binaural scenario to support separation of the target words from the background noise. The age effect observed for monaural noise was significantly reduced (although still significant) in the binaural condition. These results suggest that the effects of classroom noise on speech perception and the developmental change associated with these effects are strongly moderated by the method of sound presentation. Especially, with a simple, monaural presentation that lacks cues to spatially separate the speech signal from the noise, impairments of speech perception due to real-life environmental noise and their increase with decreasing age might be overestimated. The dominant role of spatial cues is further confirmed by the fact that, in the study of Klatte et al. [29], the impairment of speech perception due to classroom noise was considerably lower than the effects in the monaural condition, but comparable to the binaural condition of the current study (in [29], difference scores were 0.23 in children and 0.12 in adults). This was presumably because in [29], the target words were presented through a separate loudspeaker, thereby allowing spatial separation of signal and background noise.

Concerning listening comprehension, adults showed only minor disruption in the monaural condition (5%) and no significant disruption in the binaural condition. This result replicates the findings of Klatte et al. [29] and can be attributed to the adult listeners’ ability to reconstruct noise-masked elements of the speech signals using contextual cues. The age effect observed in the speech perception task was partially replicated; i.e., the second-graders were more affected by background noise when compared to third-graders and adults. However, by contrasting the results in the speech perception task, we can see that the impairment of listening comprehension performance did not differ between sound conditions (monaural vs. binaural). Furthermore, for binaural noise, children’s speech perception significantly predicted listening comprehension, whereas for monaural noise, speech perception and listening comprehension were unrelated. We may therefore conclude that, with binaural presentation of the noise, the effects on word identification allow a more valid prediction of the effects on complex listening tasks when compared to a simple monaural presentation.

Concerning the disturbance ratings, the current results replicate the findings of Klatte et al. [29] that, in view of the noise-induced performance decrements, average ratings of the children were surprisingly low. This was especially true for the monaural noise condition. Despite severe decrements in word identification, children judged this sound condition on average as “a bit disturbing.” The lowest ratings were provided by the second-graders, who showed the strongest performance impairments. Still more surprisingly, across age groups, the binaural sound scenario was judged as more disturbing, even though the speech perception decrements were much stronger in the monaural condition, and the ratings were unrelated to the actual performance decrements. Even though, with age-adequate measurement scales, children are able to provide valid ratings of general annoyance due to noise in their home and school environment [6,9,43,44], the current findings indicate that both children and adults are not aware of the detrimental effects of background noise on their listening performance. In view of this result, it is evident that teachers and researchers cannot rely on students’ ratings when evaluating the acoustic environments of classrooms.

Taken together, Exp. 1 adds to the evidence that children are more impaired than adults by noise in speech perception tasks [10,11,12,13,29], and extends this finding to listening comprehension in a realistic classroom-noise scenario. Furthermore, the results revealed that the noise effects and their interaction with age differ considerably between conditions of binaural vs. monaural presentation of the noise. In Exp. 2, we explored whether these differences hold also for cross-modal noise effects, i.e., the effects of noise on the processing of visually presented information.

## 3. Experiment 2: Effects of Classroom Noise on Visual–Verbal Short-Term Memory

In Experiment 2, the effects of a classroom-noise scenario on short-term memory for visual–verbal items were analyzed in children and adults. Verbal short-term memory is the ability to hold verbal information in an active state for ongoing cognitive processes. A standard task to assess verbal short-term memory requires immediate serial recall of sequences of 5 to 9 verbal items, such as words, digits, or easy-to-name pictures. In such tasks, participants usually employ a strategy called articulatory rehearsal, i.e., repetitive subvocal pronunciations of the list items in a sequential manner. This strategy is evident in children from about age 7, but there is considerable interindividual variation around this age [45,46].

In the current study, we used verbal short-term memory as a cognitive task for two reasons. First, the capacity of short-term memory is a predictor of children’s oral and written language acquisition, and short-term memory processes play a role in many school-based tasks, such as reading and spelling in the early grades, learning new vocabulary, mental arithmetic [47], and following the teachers’ instructions [48]. Second, short-term memory is especially susceptible to the adverse effects of noise. Many studies confirmed that performance in the serial recall task is reliably impaired by task-irrelevant background sounds (for recent reviews, see [21,49]). This “irrelevant sound effect” (ISE) is most pronounced with speech noise, but is also evoked by nonspeech sounds, such as tone sequences or instrumental music (for an overview cp. [50]). The ISE is reliable for coherent sound streams that consist of changing auditory elements emanating from a single source, e.g., fluent speech, sequences of different syllables or tones, and instrumental music. Sounds lacking these “changing-state” characteristics, such as continuous broadband noise, babble speech, or spectrally degraded speech, evoke minor or no disruption. Different theoretical explanations have been provided for the ISE. Some authors [18,51,52] attribute the effect to a diversion of attention away from the task and towards the sound (attention capture-account). Others oppose a role of attention, assuming that the effect results from specific interference between processes involved in automatic, obligatory processing of the background sound and deliberate processes involved in memorizing the verbal items [53]. Within the interference account, some authors propose serial-order retention as the mechanism of disruption [54], whereas others assume that noise—especially noise with speech—interferes with storage and processing of phonological representations [55,56].

Aiming to disentangle attention-capture from interference-by-process, a number of studies on the ISE included children. In view of children’s underdeveloped attention control [57], noise effects resulting from attention capturing should be more pronounced in children when compared to adults. However, this argument is not without problems, since, at least with noise-containing speech, stronger impairments in children may also indicate stronger speech-based interference due to less robust, immature phonological representations and/or maintenance strategies, i.e., articulatory rehearsal of the item sequence [19,21].

While the detrimental effects of irrelevant sounds on children’s verbal short-term memory were consistently reported, the findings concerning developmental change are inconsistent. Some studies reported equivalent impairments due to background speech or mixtures of nonspeech sounds with speech in 7- to 10-year-olds and adults [19,20,58,59,60,61], whereas others found stronger impairments in the children [18,19,21,62]. Two of these studies included classroom noise [20,59]. In Klatte et al. [20], children’s and adults’ serial recall performance was equally affected by background speech, but only the youngest children (first-graders) were also impaired by a mixture of classroom sounds without speech. The authors attributed the age-independent effect of background speech to specific interference with the maintenance of phonological representations and the age-dependent classroom noise effect on attentional capture. Meinhardt-Injac et al. [59] used a classroom-noise scenario with speech (bits of conversation between children and adults) and found significant and equivalent impairments of serial recall performance in 8–10-year-olds, 11–12-year-olds, and adults. Across age groups, the noise effects were unrelated to participants’ attention control, i.e., their ability to inhibit task-irrelevant information. This finding indicates that specific interference rather than attention capture is the mechanism underlying the noise-induced disruptions.

The vast majority of ISE studies have used simple, monaural presentation of irrelevant sounds. Only a few studies have included spatially spread sound sources. These studies provided evidence that the variation of the source location moderated the sounds’ disruptive effects. Buchner et al. [63] showed that the ISE evoked by nonspeech sounds (e.g., footsteps, cries of pain, and squeaking sounds) and speech (sequences of unrelated words) played through loudspeakers from different locations was most pronounced when the sound was played from the front, i.e., from a location near the visual target display to which the participants’ attention was directed. However, the source location had only a small impact on the sound-induced disruption, and the disruption evoked by speech was substantially stronger than that evoked by nonspeech sounds. These findings indicate a significant but minor role of attention capturing in the ISE in adults. Jones and Macken [64] analyzed the effects of background speech produced by six voices simultaneously. The speech was presented through loudspeakers located in a circle around the participant. The impairment of short-term memory performance was more pronounced when each voice was assigned to a separate loudspeaker (yielding six single-talker streams), when compared to assigning a mix of the six voices to each of the six loudspeakers (yielding identical streams of babble speech). Comparable results were reported in studies using dichotic vs. monaural headphone presentation of irrelevant syllables [65,66] and interpreted as evidence for specific interference through changing state speech.

In view of these findings, we might expect a stronger effect of the binaural when compared to the monaural classroom-noise scenario, through increased speech-based interference due to clearer separation of the speech streams, and/or increased attention capture due to spatially spread sound sources and changing source locations. If attention capture is the dominant source of disruption, children should be more impaired than adults, and more impaired by the binaural than the monaural noise scenario.

### 3.1. Materials and Methods

Participants: The sample consisted of 40 student volunteers (24 female), aged between 19 and 32 years (*M* = 24.0, *SD* = 2.5 years), from the University of Kaiserslautern-Landau; and 69 third- and fourth-grade children recruited from a primary school in Kaiserslautern. Due to technical issues, the data of two children had to be excluded from the analysis. The final child sample consisted of 67 children (36 female), aged between 8 years, 2 months and 10 years, 3 months (*M* = 9 years, 4 months, *SD* = 6 months). Of the children, 19 had taken part in Experiment 1. All participants were native German speakers and had normal or corrected-to-normal vision and normal hearing according to either self-reports (adults) or parental reports (children). The study was approved by the Rhineland-Palatinate school authority and by the Ethics Committee of the University of Kaiserslautern. Informed written consent was provided by the adult participants and by the children’s parents. Adults received either course credit or payment for participation (10 €).

Apparatus: Identical to Experiment 1.

Background noise: Identical to Experiment 1.

Task: The task required serial recall of sequences of monosyllabic German nouns presented pictorially. Pictures were used instead of written words in order to avoid confounding by the children’s reading abilities. Prior studies confirmed that children and adults use verbal strategies when memorizing words presented pictorially [46,67], and that participants’ strategies do not differ between pictorial and written presentation [68]. Each trial consisted of a presentation phase, a retention interval, and a recall phase. Pictures were presented one after another in a 102 × 73 mm rectangular black frame in the center of a white screen, with a presentation duration of 1500 ms and an interstimulus interval of 500 ms. A random interval between 1200 to 1800 ms passed by before the visual presentation of the first list item. The final list item was followed by a 5000 ms retention interval. The onset of the recall phase was signaled by the simultaneous re-presentation of all stimuli. The pictures were arranged at random in a fixed array of five (children) and eight (adults) black frames (see Figure 4). Participants had to reconstruct the serial order by using the mouse to click on the items in the presentation order. Clicking an item changed its shading, indicating that it had been selected. There was no time limit for responding and no possibility of error correction. After selection of the final item, participants were presented with a visual cue to start the next trial by pressing the space bar.

Both children and adults saw colored drawings representing the monosyllabic German words *Bett, Bus, Eis, Frosch, Kamm, Mond, Pilz, Schal, Schiff,* and *Zaun (bed, bus, ice, frog, comb, moon, mushroom, scarf, ship*, and *fence*). The set for the adults additionally included the items *Brief, Haus, Herz, Hut, Nuss,* and *Schwein (letter, house, heart, hat, nut,* and *pig*). Four lists of five items (drawn out of 10) were created for the children, and six lists of eight items (drawn out of 16) for the adults. Two additional versions of each list were created using random permutations of the list items.

Procedure: Both children and adults were tested in groups of 2 to 4 in a sound-attenuated booth at the University of Kaiserslautern-Landau (see Exp. 1). Adults received written instruction. Children were instructed orally by a researcher. Participants were informed that they should ignore the sounds and focus solely on the serial recall task. At the beginning of each session, the classroom scenarios were played for 4 s, followed by the presentation of all pictures used in the task. Each picture was named by a female speaker. Following the instruction, three practice trials (one per sound condition) were performed. Thereafter, children and adults completed 24 and 48 experimental trials, respectively (8 and 16 trials per sound condition). Sound conditions (silence, classroom noise—monaural, classroom noise—binaural) were varied in blocks of trials. The order of sound conditions was balanced across participants. In the sound blocks, the sound started when the participant initiated the first trial and terminated after finishing the recall phase of the final trial of the respective block. Sounds were presented via headphones at an average level of 60 dB(A). The testing session lasted about 20 min for children and 35 min for adults.

### 3.2. Results

The dependent variable was the proportion of correct scores based on the number of items recalled at the correct serial position. Proportions of correct scores with respect to age group and sound condition are depicted in Figure 5a. A two-way mixed ANOVA with sound condition (silent control, monaural, binaural) as the within-subject factor and age group (adults, children) as the between-subjects factor revealed significant main effects of sound condition (*F*(2, 210) = 8.69, *p* < 0.001, *partial η*^2^ = 0.08) and age group (*F*(1, 105) = 6.73, *p* < 0.05, *partial η*^2^ = 0.06). Bonferroni-corrected post-hoc tests revealed that performance in both noise conditions was significantly lower when compared to the silent control condition, *p* < 0.01, whereas the noise conditions did not differ (*p* = 0.99). The main effect of age group reflects better overall performance of the adults. The sound condition x age group interaction was not significant (*F* < 1), confirming comparable noise-induced disruption in adults and children. The analyses thus confirmed significant impairments of serial recall performance due to classroom noise. The effects did not differ between age groups, nor between monaural vs. binaural noise.

Aiming to assess the role of attention in the noise-induced disruption, further analyses were performed to explore whether or not participants habituated to the noise across trials. If the noise effects result from attention capture, one might expect habituation and thereby a stronger disruption in the first when compared to the final trials performed with noise [69]. For this aim, the proportion of correct scores was calculated for four consecutive blocks of two trials (children) and four trials (adults) for each sound condition. The resulting proportion correct scores are depicted in Figure 5b. A 3 × 4 × 2 mixed ANOVA with sound condition (silent control, monaural, binaural) and trial block (block 1–block 4) as within-subject factors and age group (adults, children) as the between-subjects factor was conducted. Except for sound condition and age group (reported above), neither trial block nor any interaction reached significance (all *F* < 1). The analysis thus yielded no evidence for habituation to classroom noise in children or adults.

### 3.3. Discussion

In Experiment 2, the impacts of monaural and binaural classroom-noise scenarios on verbal short-term memory were examined in children and adults. The task required serial recall of words presented pictorially. Children’s and adults’ performances were significantly and equally impaired in both noise conditions. The magnitude of the noise effects remained stable over the course of experimental trials.

The non-significant effect of age replicates the findings of Meinhardt-Injac et al. [59], who reported significant and equivalent impairments due to classroom noise in children and adults. However, in Klatte et al. [20], only the youngest children were affected by classroom noise, whereas older children and adults were unaffected. The apparent contradiction might be attributed to the kinds of classroom noise and age groups included. In both the current and the Meinhardt-Injac et al. [59] study, the classroom noise contained speech, and the youngest children were age 9 on average, whereas Klatte et al. [20] used nonspeech classroom noise and included 6- to 7-year-old first-graders. In line with the arguments provided in these studies, we propose that the detrimental effects of the classroom-noise scenario result from separate mechanisms. The spoken parts in the noise scenarios evoke specific interference with the maintenance of the list items. This mechanism may evoke stronger disruption in children whose phonological maintenance strategies are not yet fully developed and thus more prone to speech-based interference. In addition, both nonspeech sounds and speech may impair performance through attention capture. The impact of attention capture depends on the sound’s potential to grab attention (i.e., personal relevance, emotional valence, and predictability) and on the individuals’ attentional abilities. Young children are more vulnerable than older children and adults due to less developed attention control. As schooling contributes considerably to children’s development of attention control [70], preschool children and first-graders are especially vulnerable to noise-induced attention capture. Taken together, the findings of the current study add to the evidence provided by Klatte et al. [20] and Meinhardt-Injac et al. [59] that children aged around 9 years or older show adult-like impairments of short-term memory in the presence of classroom noise.

As outlined above, from both the attention capture and the interference account, one might expect stronger disruptive effects with binaural when compared to monaural presentation of the classroom-noise scenario. Following the attention capture account, especially in children, the binaural scenario should evoke stronger disruption because of its attention-grabbing quality (spatially distributed and changing sound source locations). However, contrary to expectation, the disruptive effects of the classroom-noise scenario did not differ with presentation mode, neither in children nor in adults. Furthermore, in both age groups, the detrimental effects of the noise scenarios remained stable across experimental trials—i.e., there was no evidence for habituation. These findings add to the evidence provided by Meinhardt-Injac et al. [59] and Klatte et al. [20] that diversions of attention play a minor role in the ISE, at least in children older than 8 years. This view was further confirmed in a recent study [21], demonstrating that, while 9-year-old third-graders were more impaired than adults by background speech in a verbal serial recall task, serial recall of nonverbal, visuo-spatial items was unaffected in both groups. As storage and processing of visuo-spatial items rely heavily on domain-general attentional resources [71,72], these findings strongly suggest that the impairment in the verbal task, and its interaction with age, reflect speech-based interference rather than a capture of attention.

Following the interference account, binaural sound scenarios comprising speech should evoke stronger disruption because the spatial separation of the sound sources fosters the segregation of the spoken parts into separate, single-talker streams [64]. Evidently, this mechanism was not at play in the current study. This might be due to the fact that, in the classroom-noise scenario used here, all spoken parts consisted of only two talkers simultaneously. Jones and Macken [64] showed that monaural presentations of single-talker speech and a mixture of two speakers evoked similar amounts of disruption in serial recall performance, whereas with six simultaneous speakers, the disruption was significantly reduced. Thus, concerning the current classroom-noise scenario, a clearer separation of the two speech streams through spatial cues might not further increase the interference, because the latter is already at its maximum (resembling that evoked by a single talker). On this assumption, stronger interference effects with binaural presentation should occur with noise scenarios containing more than two simultaneous talkers.

## 4. Conclusions

In the current study, the effects of a realistic classroom-noise scenario presented either monaurally or binaurally on speech perception, listening comprehension, and verbal short-term memory were investigated in primary school children and adults. In Exp. 1, across age groups, speech perception (identification of spoken words) was more im-paired by monaural than by binaural classroom noise, whereas listening comprehension (acting-out complex oral instructions) was equally impaired in both noise conditions. In both tasks, children were more affected than adults. The age effect found here is in line with a number of psychoacoustic studies documenting increasing noise-induced speech perception impairments with decreasing age [10,11,12,13] and extends these findings to a realistic noise scenario and a complex listening task that more closely reflects the requirements of children faced during classroom instruction. Disturbance ratings were unrelated to the actual performance decrements, indicating that listeners’ subjective reports do not allow a valid evaluation of adverse listening conditions. In Exp. 2, using a paradigm from the domain of the ISE, we found significant detrimental effects of the classroom-noise scenario on verbal short-term memory (serial order reconstruction of words presented pictorially), which did not differ with age or presentation format (monaural vs. binaural). Concerning the age-equivalence of the noise effect, the current finding replicates a recent study demonstrating comparable effects of classroom noise-containing speech in children aged 8 to 10 years and adults [59]. The lack of an effect of presentation format was contrary to our expectation. Especially for the children, we anticipated stronger impairments with the binaural condition due to increased attention capture through spatially separated and changing sound sources. The equivalence of the noise effects across age groups and presentation formats adds to the evidence that the ISE evoked by noise-containing speech results from specific interference between obligatory processing of the speech parts and deliberate processes involved in serial recall performance (i.e., maintenance of phonological representations), whereas attention capture plays a minor role.

Concerning age effects, children were more impaired by noise than adults in a complex listening task requiring processing of oral instructions, but equally impaired as adults in a task requiring processing of visually presented information. This indicates that, in verbal working memory tasks, children are more prone to distraction than adults when the targets and the irrelevant stimuli stem from the same sensory modality (unimodal interference) but are equally affected (or equally unaffected) as adults when the irrelevant stimuli and the targets originate from different sensory modalities (crossmodal interference, i.e., interference between visual and auditory information). It has been shown that selective attention (i.e., the ability to focus on the target stimuli and inhibit irrelevant distractors) is much more easily achieved in crossmodal paradigms when compared to unimodal paradigms. This holds especially for paradigms with visual targets and auditory distractors [73,74]. These findings have been attributed to top-down suppression of auditory distractors at a very early processing stage (at the level of the cochlea). As a consequence, age differences in attention control should be largely unrelated to auditory distraction when the relevant information is visual. This has been shown in studies including older adults [73,74,75], but as proposed by Röer et al. [60], may also hold for children. Following this view, age differences in the ISE between children and adults should emerge when the memory items are presented auditorily instead of visually. This prediction might be tested in future studies.

Concerning the impact of the noise presentation format (monaural vs. binaural), differential effects were found only in the speech perception task. The detrimental noise effect was much stronger with monaural when compared to binaural noise, and the difference between noise conditions was especially strong in the youngest (second-grade) children. These findings indicate that, with standard, monaural presentation of the maskers, the effects of noise on speech perception in real-life listening situations and the developmental change associated with speech-in-noise perception might be overestimated. Furthermore, we found that speech perception in binaural noise significantly predicted listening comprehension in binaural noise, whereas with monaural noise, speech perception and listening comprehension were unrelated. These results suggest that studying speech-in-noise perception using binaural scenarios may have the potential to allow valid predictions of detrimental noise effects on language comprehension in everyday situations. Even though the current study yielded no evidence for an effect of the noise presentation format (monaural vs. binaural) on listening comprehension and visual–verbal short-term memory, we cannot rule out that such effects exist. In the classroom-noise scenario used here, no more than two voices are presented simultaneously, and the nonspeech sounds are confined to indoor noise, disregarding noise sources from outside such as road traffic or aircraft noise. With classroom-noise scenarios containing more unexpected and novel sounds and/or more simultaneous voices, or in samples of younger children, a significantly stronger impairment with binaural when compared to monaural noise presentation might emerge.

Despite these limitations, the current study adds to the evidence concerning developmental change in the effects of noise on speech perception, listening comprehension, and short-term memory, and extends it to a realistic classroom-noise scenario. Further research is needed to find out whether and how much the use of binaural sound scenarios in experimental studies allows more valid predictions of the effects of noise in everyday, real-life learning situations.

## Figures and Tables

**Figure 1 ijerph-19-15998-f001:**
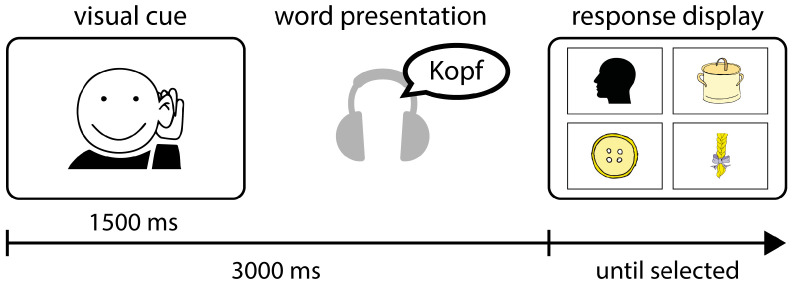
The experimental procedure for measuring speech perception with the word-to-picture matching task. The visual cue was displayed for 3000 ms, indicating the onset of the spoken target word presented over headphones. The target word was played over headphones 1500 ms after the onset of the visual cue. Thereafter, the response display was shown, which comprised four pictures, one representing the target word (here: Kopf (head)) and three representing phonologically similar distractor words (Topf (pot), Knopf (button), and Zopf (braid)).

**Figure 2 ijerph-19-15998-f002:**
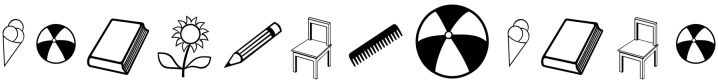
Exemplary trial from the listening comprehension task for the children. “Male ein Kreuz unter das Buch, das neben einem Stuhl liegt” (“Draw a cross under the book that is next to the chair”).

**Figure 3 ijerph-19-15998-f003:**
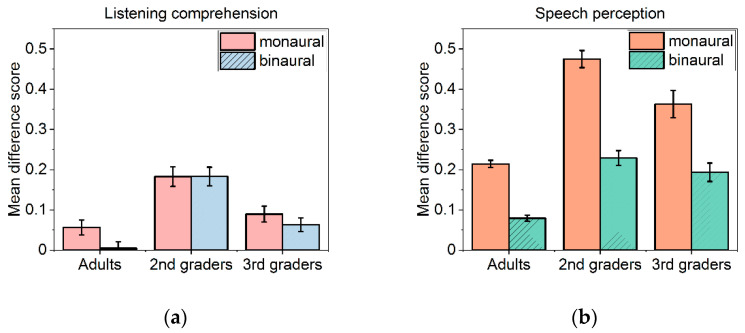
Mean difference scores for speech perception (**a**) and listening comprehension (**b**) with respect to age group and sound condition. Error bars denote standard errors of the mean.

**Figure 4 ijerph-19-15998-f004:**
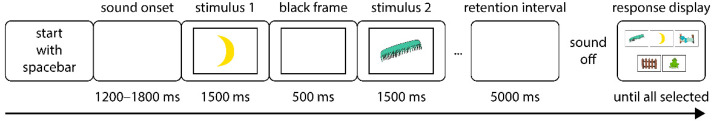
The serial recall task’s experimental procedure. Five pictures per trial were shown to the children, whereas eight pictures per trial were presented to the adults. All pictures seen in the respective trial were randomly arranged in an array of 5 (children) and 8 (adults) frames in the response display.

**Figure 5 ijerph-19-15998-f005:**
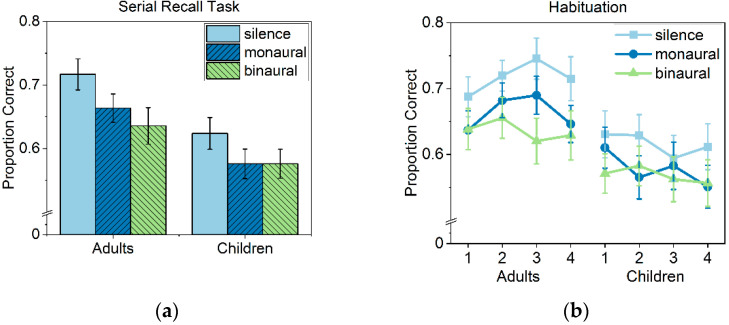
(**a**) Mean proportion correct scores as a function of age group and sound condition. (**b**) Mean proportion of correct scores as a function of age group and as a function of sound condition in four consecutive blocks of four (adults) and two (children) trials. Error bars denote standard errors of the mean.

## Data Availability

All data can be found at: https://osf.io/b9rua (accessed on 31 October 2022).

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
