# Peer review of "Differential Effects of Task-Irrelevant Monaural and Binaural Classroom Scenarios on Children’s and Adults’ Speech Perception, Listening Comprehension, and Visual–Verbal Short-Term Memory"

_ijerph, 2022, doi:10.3390/ijerph192315998_

Round 1

Reviewer 1 Report

In this paper, the effects of a realistic classroom noise scenario, presented either monaurally or binaurally, on tasks requiring the processing of auditory and visually presented information were analysed in children and adults. Results confirm the detrimental effects of noise on speech perception and cognitive performance and their interaction with age.

The methods in the paper are very well described, and the results are clearly presented.

However, there are some points that could help improve the manuscript:

1. Please provide more background information in the introductory section. Also, describe the terms "monaural" and "binaural" noise in more detail and/or give definitions, as these are key terms used throughout the manuscript.

2. Please highlight the most important contributions.

3. Please explain the term "signal-to-noise ratio".

4. Please compare the results of your experiment with the results of similar experiments (such as [26], [56], [19]...). Highlight the new conclusions you have come to.

5. There are some terms that are mentioned for the first time in the conclusion, such as "same modality", "different modalities", "crossmodal". Please explain these terms earlier, I suggest in the introduction.

Author Response

1. Please provide more background information in the introductory section. Also, describe the terms "monaural" and "binaural" noise in more detail and/or give definitions, as these are key terms used throughout the manuscript.

We extended the background information in the introduction by including a recent study on noise levels in German schools and preschools (L37 ff), and adding noise annoyance as further relevant outcome (L44f). The terms „monaural“ and „binaural“ are now briefly described in the abstract (L 18f, extended slightly from original submission) and in the manuscript on L55-56 (monaural) and L60-63, L105-106 (binaural); we also did minor changes in the method section concerning the sound scenario in order to increase clarity,  L 146-148, 172-175 f. .

2. Please highlight the most important contributions.

We did some changes in the General Discussion, so that all important contributions, and implications for future research, are now explicated. Important contributions are also summarized in the abstract.

3. Please explain the term "signal-to-noise ratio".

This is now explained in L 80-81.

4. Please compare the results of your experiment with the results of similar experiments (such as [26], [56], [19]...). Highlight the new conclusions you have come to.
Thank you for this remark. The results from Exp. 1 are discussed in relation to the findings of Klatte et al., 2010 [29] (who used similar language perception tasks), in the discussion section of Exp. 1 (L 304ff). We added a further argument concerning the findings of [29] and the current study concerning the role of spatial cues for separating signal and noise sources (L 318 ff). The results of Exp. 2 (ISE-Paradigm) are discussed now in further detail in relation to [20] and [59] (ISE studies with children using classroom noise), L 540 -581.

5. There are some terms that are mentioned for the first time in the conclusion, such as "same modality", "different modalities", "crossmodal". Please explain these terms earlier, I suggest in the introduction.

Thank you for raising this point. After consideration, we decided to explain these terms in the General discussion, where they are used for the first time (L 629-632).

Reviewer 2 Report

It's interesting research. The researchers analyze the effects of a realistic classroom noise scenario in monaural or binaural fashion on tasks that require the processing of auditory and visually presented information.

The final infant sample consisted of 67 children. It would have been interesting to expand the sample.

The Discussion section is well founded.

I consider that it is a well-structured article, with coherence between objectives and results.

Author Response

Thanks to Reviewer 2 for the positive evaluation of our study. We fully agree that it might be interesting to increase the size of the children sample. Especially, inlcuding a sample of younger children (first-graders) would help to explore the role of attention capture in the disruptive effect of classroom noise on short-term memory, as we outline in the Discussion of Exp. 1 (L 540 ff) and in the General Discussion (L 661-663).

Reviewer 3 Report

As a background piece of general information, the reviewer brings in a quote from the Finnish Adapteo Group,

"A computer’s noise level is 30-50 decibels, conversation makes 50-70 decibels and traffic will cause a sound of 70 to 85 decibels. The ear’s pain threshold is 125 decibels. There is a risk of hearing loss when the noise level rises to 85 decibels for eight hours repeatedly. According to measurements made in (Finnish) schools, the World Health Organization (WHO)recommendations may be exceeded even in an empty classroom. The noise is caused by air conditioning, traffic and activities in other parts of school facilities. During studying, the level of noise in the classroom depends on the size of the group and their ways of working.

The noise level in a school can rise momentarily over 90 decibels

The average noise level for eight hours is typically 70-80 decibels in kindergartens and schools. In certain situations and spaces, such as lobbies or hallways, cafeterias, moving from one place to another and during free play, the noise level can temporarily rise to over 90 decibels.

Mirka Hintsanen, professor of psychology at the University of Oulu, Finland, reminds in her article about noise research that adults find noise less disturbing than children, so it might be difficult for adults to understand how disturbing noise can be for children. Children can have very little influence on their sound environment at school and it is usually not possible for them to move to another, quieter room to do tasks which require concentration. " (Adapteo.com, accessed 14 Nov. 2022)

Two questions are being posed for the authors who have done a meticulous and exceptional report on their investigation. 

Question 1, the above quotation reminds one of the specific nature of measuring noise over time exposure by the concept of equivalent noise level. It also reminds us (if the same is also true in Germany or most countries) that children are stationed in one specific classroom instead of the university students who moved from classrooms to classrooms for their classes hence are somewhat 'captive' of the noise environment of their classroom to being with.  If the above assumptions are fair, the researchers are asked to respond to the first question, why hasn't  the research addressed the concept of equivalent noise level over a typical school day which represents that the children's binaural or monaural experience have in fact been pre-conditioned due to repetitive daily experience. How does this preconditioning factor into the research when compared with the university students who do not have a 'home classroom'?

The second question refers to the authors' discussion and conclusion which referred to the works by Klatte, and Reinhardt-Injac, what kind of contribution to knowledge have been produced by the authors? Can they elaborate on the significance of their findings?

Author Response

Two questions are being posed for the authors who have done a meticulous and exceptional report on their investigation. 

Question 1, the above quotation reminds one of the specific nature of measuring noise over time exposure by the concept of equivalent noise level. It also reminds us (if the same is also true in Germany or most countries) that children are stationed in one specific classroom instead of the university students who moved from classrooms to classrooms for their classes hence are somewhat 'captive' of the noise environment of their classroom to being with.  If the above assumptions are fair, the researchers are asked to respond to the first question, why hasn't  the research addressed the concept of equivalent noise level over a typical school day which represents that the children's binaural or monaural experience have in fact been pre-conditioned due to repetitive daily experience. How does this preconditioning factor into the research when compared with the university students who do not have a 'home classroom'?

Thank you for the comment. The values reported were indeed equivalent continuous noise levels for daily activities in schools. This has been clarified better now, and we added a further reference on noise levels in German classrooms (see lines 37-41). We are not able to consider preconditioning, as we don’t know of any respective studies. However, we assume that university students are also used to classroom-like noise, as they are daily instructed in seminar and lecture rooms [Ricciardi, P., & Buratti, C. (2018). Environmental quality of university classrooms: Subjective and objective evaluation of the thermal, acoustic, and lighting comfort conditions. Building and Environment, 127, 23-36].  

The second question refers to the authors' discussion and conclusion which referred to the works by Klatte, and Reinhardt-Injac, what kind of contribution to knowledge have been produced by the authors? Can they elaborate on the significance of their findings?

The results of Exp. 2 (ISE-Paradigm) are discussed now in further detail in relation to Klatte et al. (2010) [20] and Meinhardt-Injac et al., (2022) [59], L 540 -581. Furthermore, we did some changes in the General Discussion, so that all significant contributions, and implications for future research, are now explicated more clearly. Main findings and conclusions are also summarized in the abstract.